# Hypernetwork-based Meta-Learning
# for Low-Rank Physics-Informed Neural Networks

**Woojin Cho**[†] **Kookjin Lee**[‡*] **Donsub Rim**[§] **Noseong Park**[†*]

[†] Yonsei University
[‡] Arizona State University
[§] Washington University in St. Louis
snowmoon@yonsei.ac.kr, kookjin.lee@asu.edu,
rim@wustl.edu, noseong@yonsei.ac.kr

## Abstract

In various engineering and applied science applications, repetitive numerical simulations of partial differential equations (PDEs) for varying input parameters are often required (e.g., aircraft shape optimization over many design parameters) and solvers are required to perform rapid execution. In this study, we suggest a path that potentially opens up a possibility for physics-informed neural networks (PINNs), emerging deep-learning-based solvers, to be considered as one such solver. Although PINNs have pioneered a proper integration of deep-learning and scientific computing, they require repetitive time-consuming training of neural networks, which is not suitable for *many-query* scenarios. To address this issue, we propose lightweight low-rank PINNs containing only hundreds of model parameters and an associated hypernetwork-based meta-learning algorithm, which allow efficient solution approximations for varying PDE input parameters. Moreover, we show that the proposed method is effective in overcoming a challenging issue, known as "failure modes" of PINNs.

## 1 Introduction

Physics-informed neural networks (PINNs) [1] are a particular class of coordinate-based multi-layer perceptrons (MLPs), also known as implicit neural representations (INRs), to numerically approximate solutions of partial differential equations (PDEs). That is, PINNs are taking spatiotemporal coordinates $(\boldsymbol{x}, t)$ as an input and predict PDE solutions evaluated at the coordinates $u_\Theta(\boldsymbol{x}, t)$ and are trained by minimizing (implicit) PDE residual loss and data matching loss at initial and boundary conditions. PINNs have been successfully applied to many different important applications in computational science and engineering domain: computational fluid dynamics [2, 3], cardiac electrophysiology simulation [4], material science [5], and photonics [6], to name a few.

PINNs are, however, sharing the same weakness with coordinate-based MLPs (or INRs), which hinders the application of PINNs/INRs to more diverse applications; for a new data instance (e.g., a new PDE for PINNs or a new image for INRs), training a new neural network (typically from scratch) is required. Thus, using PINNs to solve PDEs (particularly, in parameterized PDE settings) is usually computationally demanding, and this burden precludes the application of PINNs to important scenarios that involve *many queries* in nature as these scenarios require the parameterized PDE models to be simulated thousands of times (e.g., design optimization, uncertainty propagation), i.e., requiring PDE solutions $u(\boldsymbol{x}, t; \boldsymbol{\mu})$ at many PDE parameter settings $\{\boldsymbol{\mu}^{(i)}\}_{i=1}^{N_\mu}$ with very large $N_\mu$.

---

[*]Co-corresponding authors

37th Conference on Neural Information Processing Systems (NeurIPS 2023).

To mitigate the above described issue, we propose i) a low-rank structured neural network architecture for PINNs, denoted as low-rank PINNs (LR-PINNs), ii) an efficient rank-revealing training algorithm, which adaptively adjust ranks of LR-PINNs for varying PDE inputs, and iii) a two-phase procedure (offline training/online testing) for handling *many-query* scenarios. This study is inspired by the observations from the studies of numerical PDE solvers [7–11] stating that numerical solutions of parametric PDEs can be often approximated in a low-rank matrix or tensor format with reduced computational/memory requirements. In particular, the proposed approach adopts the computational formalism used in reduced-order modeling (ROM) [12, 13], one of the most dominant approaches in solving parameteric PDEs, which we will further elaborate in Section 3.

In essence, LR-PINNs represent the weight of some internal layers as a low-rank matrix format. Specifically, we employ a singular-value-decomposition (SVD)-like matrix decomposition, i.e., a linear combination of rank-1 matrices: the weight of the $l$-th layer is $W^l = \sum_{i=1}^r s_i^l \boldsymbol{u}_i^l \boldsymbol{v}_i^{l\mathsf{T}}$ with the rank $r = \min(n_l, n_{l+1})$, where $W^l \in \mathbb{R}^{n_{l+1} \times n_l}$, $\boldsymbol{u}_i^l \in \mathbb{R}^{n_{l+1}}$, $\boldsymbol{v}_i^l \in \mathbb{R}^{n_l}$, and $s_i^l \in \mathbb{R}$. The ranks of the internal layers, however, typically are not known a priori. To address this issue, we devise a novel hypernetwork-based neural network architecture, where the rank-structure depending on the PDE parameters $\boldsymbol{\mu}$ is learned via training. In short, the proposed architecture consists of i) a *lightweight* hypernetwork module and ii) a low-rank solution network module; the hypernetwork takes in the PDE parameters and produces the coefficients of the rank-1 series expansion (i.e., $\boldsymbol{s}(\boldsymbol{\mu}) = f^{\text{hyper}}(\boldsymbol{\mu})$). The low-rank solution network module i) takes in the spatiotemporal coordinates $(\boldsymbol{x}, t)$, ii) takes the forward pass through the linear layers with the weights $W^l(\boldsymbol{\mu}) = \sum_{i=1}^r s_i^l(\boldsymbol{\mu}) \boldsymbol{u}_i^l \boldsymbol{v}_i^{l\mathsf{T}}$, of which $s_i(\boldsymbol{\mu})$ comes from the hypernetwork, and iii) produces the prediction $u_\Theta(\boldsymbol{x}, t; \boldsymbol{\mu})$. Then, the training is performed via minimizing the PINN loss, i.e., a part of the PINN loss is the PDE residual loss, $\|\mathcal{R}(u_\Theta(x, t; \boldsymbol{\mu}); \boldsymbol{\mu})\|_2$, where $\mathcal{R}(\cdot, \cdot; \boldsymbol{\mu})$ denotes the parameterized PDE residual operator.

We show the efficacy and the efficiency of our proposed method for solving some of the most fundamental parameterized PDEs, called *convection-diffusion-reaction equations*, and *Helmholtz equations*. Our contributions include:

1. We employ a low-rank neural network for PINNs after identifying three research challenges to address.
2. We develop a hypernetwork-based framework for solving parameterized PDEs, which computes solutions in a rank-adaptive way for varying PDE parameters.
3. We demonstrate that the proposed method resolves the "*failure modes*" of PINNs.
4. We also demonstrate that our method outperforms baselines in terms of accuracy and speed.

## 2 Naïve low-rank PINNs

Let us being by formally defining LR-PINNs and attempt to answer relevant research questions. LR-PINNs are a class of PINNs that has hidden fully-connected layers (FC) represented as a low-rank weight matrix. We denote this intermediate layer as LR-FC: the $l$-th hidden layer is defined such that

$$\boldsymbol{h}^{l+1} = \text{LR-FC}^l(\boldsymbol{h}^l) \quad \Leftrightarrow \quad \boldsymbol{h}^{l+1} = U_r^l(\Sigma_r^l(V_r^{l\mathsf{T}} \boldsymbol{h}^l)) + \boldsymbol{b}^l, \tag{1}$$

where $U_r^l \in \mathbb{R}^{n_{l+1} \times r}$ and $V_r^l \in \mathbb{R}^{n_l \times r}$ denote full column-rank matrices (i.e., rank $r \ll n_l, n_{l+1}$) containing a set of orthogonal basis vectors, and $\Sigma_r^l \in \mathbb{R}^{r \times r}$ is a diagonal matrix, $\Sigma_r^l = \text{diag}\left(\boldsymbol{s}_r^l\right)$ with $\boldsymbol{s}_r^l \in \mathbb{R}^r$.

**Memory efficiency:** LR-PINNs with a rank of $r$ and $L$ hidden layers require $O((2n_l + 1)rL)$ memory as opposed to $O(n_l^2 L)$ required by regular PINNs.

**Computational efficiency:** The forward/backward pass of LR-PINNs can be computed efficiently by utilizing a factored representation of the weights. To simplify the presentation, we describe only the forward pass computation; the forward pass is equivalent to perform three small matrix-vector products (MVPs) in sequence as indicated by the parentheses in Eq. (1).

**Challenges:** Representing the weights of hidden layers itself is straightforward and indeed has been studied actively in many different fields of deep learning, e.g., NLP [14, 15]. However, those approaches typically assume that there exist pre-trained models and approximate the model weights by running the truncated SVD algorithm. Our approach is different from these approaches in that we

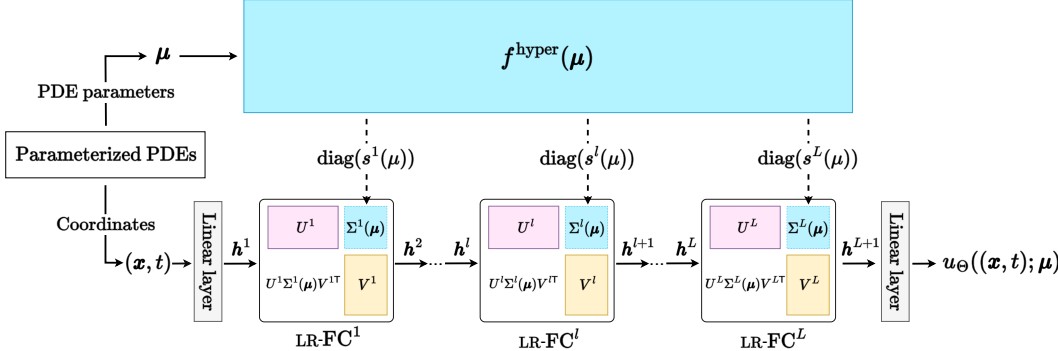

Figure 1: The architecture of Hyper-LR-PINN consisting of i) the hypernetwork generating model parameters (i.e., diagonal elements) of LR-PINN and ii) LR-PINN approximating solutions.

attempt to reveal the ranks of internal layers as the training proceeds, which brings unique challenges. These challenges can be summarized with some research questions, which include: **C1**) "should we make all parameters learnable (i.e., $(U^l, V^l, \Sigma^l)$)?", **C2**) "how can we determine the ranks of each layer separately, and also adaptively for varying $\boldsymbol{\mu}$?", and **C3**) "can we utilize a low-rank structure to avoid expensive and repetitive training of PINNs for every single new $\boldsymbol{\mu}$ instances?". In the following, we address these questions by proposing a novel neural network architecture.

## 3 Hyper-LR-PINNs: hypernetwork-based meta-learning low-rank PINNs

We propose a novel neural network architecture based on a hypernetwork and an associated training algorithm, which address the predescribed challenges. To distinguish them, we hereinafter call LR-PINN with (resp. w/o) our proposed hypernetwork as "Hyper-LR-PINN" (resp. "Naïve-LR-PINN").

**Design goals:** Here we attempt to resolve all challenges by setting up several design goals that are inspired from our domain knowledge in the field (and also from some preliminary results that can be found in Appendix D):

1. build a *single set* of basis vectors $U^l$ and $V^l$, preferably as *orthogonal* as possible, that perform well over a range of PDE parameters,
2. build LR-PINNs with an *adaptive and layer-wise rank structure* that depends on PDE parameters (e.g., a higher rank for a higher convective PDE) and,
3. make *only the diagonal elements*, denoted $\boldsymbol{s}_r^l$ in Eq. (1), *learnable* to achieve high efficiency once a proper set of basis vectors and the rank structure are identified.

Our design largely follows the principles of ROMs, where the expensive computation is offloaded to an *offline* phase to build a cheap surrogate model that can perform an efficient computation in an *online* phase for a test set. To make an analogy, consider parameterized dynamical systems (which may arise in semi-discretization of time-dependent PDEs): $\frac{\mathrm{d}\boldsymbol{u}(t;\boldsymbol{\mu})}{\mathrm{d}t} = \boldsymbol{f}(\boldsymbol{u}(t;\boldsymbol{\mu});\boldsymbol{\mu})$, where $\boldsymbol{u} \in \mathbb{R}^N$. In the offline phase, the method seeks a low-dimensional linear trial basis where the reduced representation of the solution lie on, which is achieved by performing high-fidelity simulations on a set of PDE parameter instances $\{\boldsymbol{\mu}^{(i)}\}_{i=1}^{n_{\mathrm{train}}}$ and constructing a trial linear subspace $\Psi := \mathrm{range}(\Psi_p)$ with $\Psi_p = [\psi_1, \cdots, \psi_p] \in \mathbb{R}^{N \times p}$ from the solution snapshots collected from the high-fidelity simulations. In the online phase, the solutions at a set of test PDE parameter instances $\{\boldsymbol{\mu}^{(i)}\}_{i=1}^{n_{\mathrm{test}}}$ are approximated as $\boldsymbol{u}(t,\boldsymbol{\mu}) \approx \Psi_p \boldsymbol{c}(t,\boldsymbol{\mu})$ with $\boldsymbol{c} \in \mathbb{R}^p$, and a low-dimensional surrogate problem is derived as $\frac{\mathrm{d}\boldsymbol{c}(t;\boldsymbol{\mu})}{\mathrm{d}t} = \Psi_p^{\mathsf{T}} \boldsymbol{f}(\Psi_p \boldsymbol{c}(t;\boldsymbol{\mu});\boldsymbol{\mu}) = \hat{\boldsymbol{f}}(\boldsymbol{c}(t;\boldsymbol{\mu});\boldsymbol{\mu}) \in \mathbb{R}^p$, which can be rapidly solved, while not losing too much accuracy. See Appendix C for an illustrative explanation and details of ROMs.

Taking a cue from the ROM principles, we design our model to operate on a common set of basis vectors $\{U_r^l\}$ and $\{V_r^l\}$, which are obtained during the offline phase (analogous to $\Psi_p$ in ROMs) and update only the diagonal elements $\{\boldsymbol{s}_r^l\}$ during the online phase (analogous to $\boldsymbol{c}$ in ROMs). Now, we elaborate our network design and the associated two-phase algorithm. For the connection to the ROM, which explains details on the context/query sets, can be found in Appendix C.

## 3.1 Hypernetwork-based neural network architecture

The proposed framework has two computational paths: a path for the hypernetwork and a path for LR-PINN (Figure 1). The hypernetwork path reads the PDE parameter $\boldsymbol{\mu}$ and outputs the diagonal elements $\{\boldsymbol{s}^l\}_{l=1}^L$ of LR-FCs. The LR-PINN path reads $(\boldsymbol{x},t)$ and the output of the hypernetwork, and outputs the approximated solution $u_\Theta$ at $(\boldsymbol{x},t;\boldsymbol{\mu})$, which can be written as follows:

$$u_\Theta((\boldsymbol{x},t);\boldsymbol{\mu}) = u_\Theta((\boldsymbol{x},t); f^{\text{hyper}}(\boldsymbol{\mu})),$$

where $f^{\text{hyper}}(\boldsymbol{\mu})$ denotes the hypernetwork such that $\{\boldsymbol{s}^l(\boldsymbol{\mu})\}_{l=1}^L = f^{\text{hyper}}(\boldsymbol{\mu})$. We denote $\boldsymbol{s}$ as a function of $\boldsymbol{\mu}$ to make it explicit that it is dependent on $\boldsymbol{\mu}$. The internals of LR-PINN can be described as with $\boldsymbol{h}^0 = [\boldsymbol{x},t]^\mathsf{T}$

$$\boldsymbol{h}^1 = \sigma(W^0\boldsymbol{h}^0 + \boldsymbol{b}^0),$$
$$\boldsymbol{h}^{l+1} = \sigma(U^l(\Sigma^l(\boldsymbol{\mu})(V^{l\mathsf{T}}\boldsymbol{h}^l)) + \boldsymbol{b}^l), l = 1,\ldots,L,$$
$$u_\Theta((\boldsymbol{x},t);\boldsymbol{\mu}) = \sigma(W^{L+1}\boldsymbol{h}^{L+1} + \boldsymbol{b}^{L+1}),$$

where $\Sigma^l(\boldsymbol{\mu}) = \text{diag}(\boldsymbol{s}^l(\boldsymbol{\mu}))$. The hypernetwork can be described as the following initial embedding layer, where $\boldsymbol{e}^0 = \boldsymbol{\mu}$, followed by an output layer:

$$\boldsymbol{e}^m = \sigma(W^{\text{emb},m}\boldsymbol{e}^{m-1} + \boldsymbol{b}^{\text{emb},m}), \quad m = 1,\ldots,M,$$
$$\boldsymbol{s}^l(\boldsymbol{\mu}) = \text{ReLU}(W^{\text{hyper},l}\boldsymbol{e}^M + \boldsymbol{b}^{\text{hyper},l}), \quad l = 1,\ldots,L,$$

where ReLU is employed to automatically truncate the negative values so that the adaptive rank structure for varying PDE parameters can be revealed (i.e., the number of non-zeros (NNZs)[2] in $\boldsymbol{s}^l(\boldsymbol{\mu})$ varies depending on $\boldsymbol{\mu}$).

## 3.2 Two-phase training algorithm

Along with the framework, we present the proposed two-phase training algorithm. Phase 1 is for learning the common set of basis vectors and the hypernetwork and Phase 2 is for fine-tuning the network for a specific set of test PDE parameters. Table 1 shows the sets of model parameters that are being trained in each phase. (See Appendix E for the formal algorithm.)

In Phase 1, we train the hypernetwork and the LR-PINN jointly on a set of collocation points that are collected for varying PDE parameters. Through the computational procedure described in Section 3.1, the approximated solutions at the collocation points are produced $u_\Theta((\boldsymbol{x}_j,t_j);\boldsymbol{\mu}^{(i)})$. Then, as in regular PINNs, the PDE residual loss and the data matching loss can be computed. The small difference is that the PDE operator, $\mathcal{F}$, for the residual loss is also parameterized such that $\mathcal{F}(u_\Theta((\boldsymbol{x}_j,t_j);\boldsymbol{\mu}^{(i)});\boldsymbol{\mu}^{(i)})$. As we wish to obtain basis vectors that are close to orthogonal, we add the following orthogonality constraint based on the Frobenius norm to the PINN loss [16]:

$$w_1\|U^{l\mathsf{T}}U^l - I\|_F^2 + w_2\|V^{l\mathsf{T}}V^l - I\|_F^2, \tag{2}$$

where $w_1$ and $w_2$ are penalty weights. (See Appendix J.)

In Phase 2, we continue training LR-PINN for approximating the solutions of a target PDE parameter configuration. We i) fix the weights, the biases, and the set of basis vectors of LR-FC obtained from Phase 1 , ii) convert the diagonal elements to a set of learnable parameters after initializing them with the values from the hypernetwork, and iii) detach the hypernetwork. Thus, only the trainable parameters from this point are the set of diagonal elements, first and last linear layers. The hypernetwork-initialized diagonal elements serve as a good starting point in (stochastic) gradient update optimizers (i.e., require less number of epochs). Moreover, significant computational savings can be achieved in the gradient update steps as only the diagonal elements are updated (i.e., $\boldsymbol{s}_{l+1} \leftarrow \boldsymbol{s}_l + \eta\nabla_{\boldsymbol{s}}L$ instead of $\Theta_{l+1} \leftarrow \Theta_l + \eta\nabla_\Theta L$).

## 4 Experiments

We demonstrate that our proposed method significantly outperforms baselines on the 1-dimensional/2-dimensional PDE benchmarks that are known to be very challenging for PINNs to learn [17, 18]. We report the average accuracy and refer readers to Appendix U for the std. dev. of accuracy after 3 runs.

---

[2]The NNZs is not equal to the rank as the learned basis is only soft-constrained to be orthogonal (Eq. (2)). Nevertheless, to avoid notation overloading, we denote NNZs as the rank in the following.

Table 1: Learnable parameters in each phase

| Phase 1 | $\{(U^l, V^l, \boldsymbol{b}^l)\}_{l=1}^L, W^0, W^{L+1}, \boldsymbol{b}^0, \boldsymbol{b}^{L+1}$      (LR-PINN), 
 $\{(W^{\mathrm{emb},m}, \boldsymbol{b}^{\mathrm{emb},m})\}_{m=1}^M, \{(W^{\mathrm{hyper},l}, \boldsymbol{b}^{\mathrm{hyper},l})\}_{l=1}^L$ (hypernetwork), |
|---|---|
| Phase 2 | $\{\boldsymbol{s}^l\}_{l=1}^L, W^0, W^{L+1}, \boldsymbol{b}^0, \boldsymbol{b}^{L+1}$      (LR-PINN) |

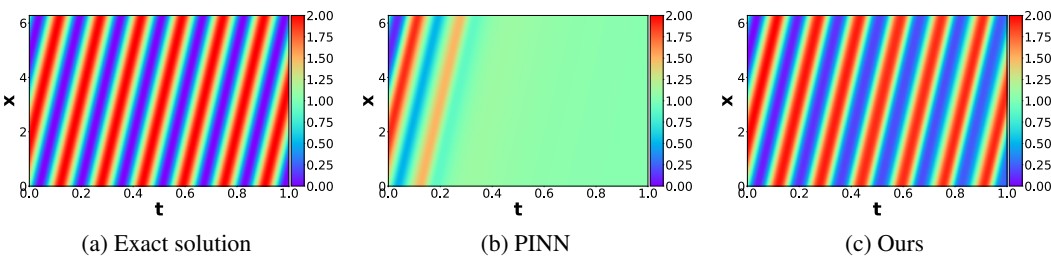

(a) Exact solution            (b) PINN            (c) Ours

Figure 2: [Convection equation] Solution snapshots for $\beta = 40$

**Baselines for comparison:** Along with the vanilla PINN [1], our baselines include several variants. PINN-R [19] denotes a model that adds skip-connections to PINN. PINN-S2S [18] denotes a method that uniformly segments the temporal domain and proceeds by training each segment one by one in a temporal order. PINN-P denotes a method that directly extends PINNs to take $(\boldsymbol{x}, t, \boldsymbol{\mu})$ as input and infer $u_\theta(\boldsymbol{x}, t, \boldsymbol{\mu})$, i.e., $\boldsymbol{\mu}$ is being treated as a coordinate in the parameter domain.

We also apply various meta-learning algorithms to Naïve-LR-PINN: model-agnostic meta learning (MAML) [20] and Reptile [21] — recall that Naïve-LR-PINN means that LR-PINN without our proposed hypernetwork-based meta-learning and therefore, MAML and Reptile on top of Naïve-LR-PINN can be compared to Hyper-LR-PINN. In the parameterized PDE setting, we can define a task, $\tau^{(i)}$, as a specific setting of the PDE parameters, $\boldsymbol{\mu}^{(i)}$. Both MAML and Reptile seek initial weights of a PINN, which can serve as a good starting point for gradient-based optimizers when a solution of a new unseen PDE parameter setting is sought. See Appendix G for details. For reproducibility, we refer readers to Appendix F, including hyperparameter configuration and software/hardware environments.

**Evaluation metrics:** Given the $i$-th PDE parameter instance $\boldsymbol{\mu}^{(i)}$, the ground-truth solution evaluated at the set of test collocation points can be defined collectively as $\boldsymbol{u}^{(i)} = [u(x_1, t_1; \boldsymbol{\mu}^{(i)}), \ldots, u(x_N, t_N; \boldsymbol{\mu}^{(i)})]^\mathsf{T}$ and likewise for PINNs as $\boldsymbol{u}_\theta^{(i)}$. Then the absolute error and the relative error can be defined as $\frac{1}{N}\|\boldsymbol{u}^{(i)} - \boldsymbol{u}_\theta^{(i)}\|_1$ and $\|\boldsymbol{u}^{(i)} - \boldsymbol{u}_\theta^{(i)}\|_2 / \|\boldsymbol{u}^{(i)}\|_2$, respectively. In Appendix N, we measure the performance on more metric: max error and explained variance score.

### 4.1 Benchmark parameterized PDEs

**1D PDEs:** For our first benchmark parameterized PDEs, we consider the following parameterized convection-diffusion-reaction (CDR) equation:

$$u_t + \beta u_x - \nu u_{xx} - \rho u(1 - u) = 0, \quad x \in \Omega, \ t \in [0, T],$$

where the equation describes how the state variable $u$ evolves under convective (the second term), diffusive (the third term), and reactive (the fourth term) phenomena.[3] The triplet, $\boldsymbol{\mu} = (\beta, \nu, \rho)$, defines the characteristics of the CDR equations: how strong convective/diffusive/reactive the equation is, respectively. A particular choice of $\boldsymbol{\mu}$ leads to a realization of a specific type of CDR processes.

There is a set of $\boldsymbol{\mu}$ which makes training PINNs very challenging, known as "failure modes" [18]: i) convection equations with high convective terms ($\beta \geq 30$) and ii) reaction(-diffusion) equations with high reactive terms ($\rho \geq 5$). We demonstrate that our method does not require specific-PDE-dedicated algorithms (e.g., [18]) while producing comparable/better accuracy in low rank.

---

[3]Note that we consider the Fisher's form $\rho u(1 - u)$ as the reaction term following [18].

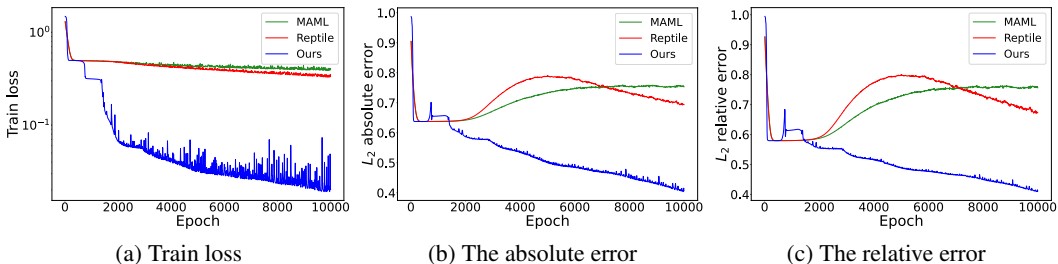

|                |                |                |
| :------------: | :------------: | :------------: |
| (a) Train loss | (b) The absolute error | (c) The relative error |

Figure 3: [Convection equation] Per epoch averaged train loss and test errors for $\beta \in [30, 40]$ for Phase 1 (meta-learning phase). We refer readers to Appendix O for Phase 2 loss curve.

Table 2: [Convection equation] The absolute and relative errors of the solutions of convection equations with $\beta = \{30, 35, 40\}$. We apply the curriculum learning proposed in [18], MAML, and Reptile to Naïve-LR-PINN. Therefore, we focus on comparing various Naïve-LR-PINN-based enhancements and our Hyper-LR-PINN. See Appendix N for other omitted tables.

| | | [w/o] Pre-training | | | | [w] Pre-training | | | | | | | |
| :-: | :-: | :-: | :-: | :-: | :-: | :-: | :-: | :-: | :-: | :-: | :-: | :-: | :-: |
| $\beta$ | Rank | Naïve-LR-PINN | | Curriculum learning | | MAML | | Reptile | | Hyper-LR-PINN (Full rank) | | Hyper-LR-PINN (Adaptive rank) | |
| | | Abs. err. | Rel. err. | Abs. err. | Rel. err. | Abs. err. | Rel. err. | Abs. err. | Rel. err. | Abs. err. | Rel. err. | Abs. err. | Rel. err. |
| | 10 | 0.5617 | 0.5344 | 0.4117 | 0.4098 | 0.6757 | 0.6294 | 0.5893 | 0.5551 | | | | |
| | 20 | 0.5501 | 0.5253 | 0.4023 | 0.4005 | 0.6836 | 0.6452 | 0.6144 | 0.5779 | | | | |
| 30 | 30 | 0.5327 | 0.5126 | 0.4233 | 0.4204 | 0.5781 | 0.5451 | 0.6048 | 0.5704 | 0.0360 | 0.0379 | 0.0375 | 0.0389 |
| | 40 | 0.5257 | 0.5076 | 0.3746 | 0.3744 | 0.5848 | 0.5515 | 0.5757 | 0.5442 | | | | |
| | 50 | 0.5327 | 0.5126 | 0.4152 | 0.4127 | 0.5898 | 0.5562 | 0.5817 | 0.5496 | | | | |
| | 10 | 0.5663 | 0.5357 | 0.5825 | 0.5465 | 0.6663 | 0.6213 | 0.5786 | 0.5446 | | | | |
| | 20 | 0.5675 | 0.5369 | 0.6120 | 0.5673 | 0.6814 | 0.6433 | 0.5971 | 0.5606 | | | | |
| 35 | 30 | 0.6081 | 0.5670 | 0.5864 | 0.5503 | 0.5819 | 0.5466 | 0.5866 | 0.5506 | 0.0428 | 0.0443 | 0.0448 | 0.0461 |
| | 40 | 0.5477 | 0.5227 | 0.5954 | 0.5548 | 0.5809 | 0.5462 | 0.5773 | 0.5435 | | | | |
| | 50 | 0.5449 | 0.5208 | 0.6010 | 0.5619 | 0.5870 | 0.5514 | 0.5731 | 0.5404 | | | | |
| | 10 | 0.5974 | 0.5632 | 0.5978 | 0.5611 | 0.6789 | 0.6446 | 0.5992 | 0.5632 | | | | |
| | 20 | 0.5890 | 0.5563 | 0.6274 | 0.5820 | 0.7008 | 0.6801 | 0.6189 | 0.5853 | | | | |
| 40 | 30 | 0.6142 | 0.5724 | 0.6011 | 0.5652 | 0.6072 | 0.5700 | 0.6126 | 0.5810 | 0.0603 | 0.0655 | 0.0656 | 0.0722 |
| | 40 | 0.5560 | 0.5293 | 0.6126 | 0.5715 | 0.6149 | 0.5832 | 0.6004 | 0.5638 | | | | |
| | 50 | 0.6161 | 0.5855 | 0.6130 | 0.5757 | 0.6146 | 0.5799 | 0.6007 | 0.5645 | | | | |

**2D PDEs:** As the second set of benchmarks, we consider the 2-dim parameterized Helmholtz equation:

$$u_{xx} + u_{yy} + k^2 u - q(x, y; a_1, a_2) = 0,$$

where $q(x, y; a_1, a_2)$ denotes a specific parameterized forcing term and the solution $u$ can be calculated analytically (See Appendix T). As observed in the failure modes of convection equations (high convective terms), certain choices of the parameters in the forcing term, i.e., $a_1, a_2$, make the training of PINNs challenging as the solutions become highly oscillatory.

### 4.2 Experimental results

#### 4.2.1 Performance on the failure mode of the CDR equation

**Solution accuracy:** As studied in prior work, vanilla PINNs fail to approximate solutions exhibiting either highly oscillatory (due to high convection) or sharp transient (due to high reaction) behaviors. Here, we present the results of convection equations and leave the results on reaction(-diffusion) equations in Appendix N. For convection equations[4], failures typically occur with high $\beta$, e.g., $\beta \geq 30$. Table 2 reports the results of all considered models trained on $\beta \in [30, 40]$[5] and essentially shows that Hyper-LR-PINN (Figure 2(c)) is the only low-rank method that can resolve the failure-mode and there are only marginal decreases in accuracy compared to Hyper-LR-PINN in full rank. For reaction(-diffusion) equations, we observe similar results, Hyper-LR-PINN outperforms the

---

[4]Initial condition: $1 + \sin(x)$ and boundary condition: periodic

[5]Following [18], we consider only the coefficients in the natural numbers (e.g., $\beta \in \mathbb{N}_+$) and [30,40] indicates the set $\{30, 31, \ldots, 40\}$.

Table 3: Comparisons of model size

| Model | Naïve-LR-PINN | | | | | Ours | PINN |
|---|---|---|---|---|---|---|---|
| Rank | 10 | 20 | 30 | 40 | 50 | Adaptive | - |
| # Parameters | 381 | 411 | 441 | 471 | 501 | $\sim$351 | 10,401 |

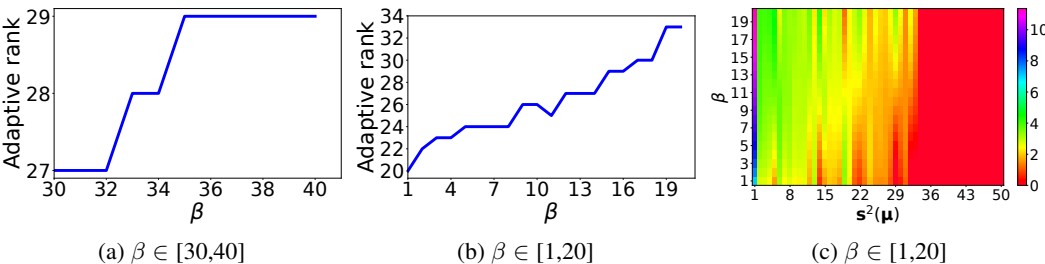

(a) $\beta \in [30, 40]$      (b) $\beta \in [1, 20]$      (c) $\beta \in [1, 20]$

Figure 4: Adaptive rank on convection equation (the left and the middle panels). The magnitude of the learned diagonal elements $\boldsymbol{s}^2$ of the second hidden layer for varying $\beta \in [1, 20]$ (the right panel).

best baseline by more than an order of magnitude (See Appendix N). Moreover, we report that Hyper-LR-PINN outperforms in most cases to the baselines that operate in "full-rank": meta-learning methods (MAML, Reptile), PINN-S2S, and PINN-P (Appendix M).

**Loss curves:** Figure 3 depicts two curves of train loss and test errors as the meta-learning algorithms proceed (MAML, Reptile, and ours). As opposed to the optimization-based meta-learning algorithms, which tend to learn meta-initial weights that perform well "on average" over randomly sampled training tasks, our hypernetwork-based method minimizes the loss for each individual task simultaneously. Loss curves for Phase 2 are reported in Appendix O, which essentially show that the baseline meta-learners do not provide good initializations.

**Rank structure:** Table 3 compares the number of trainable model parameters for Naïve-LR-PINN, PINN, and our method. In our model, each hidden layer has a different rank structure, leading to 351 trainable parameters, which is about ($\times$30) smaller than that of the vanilla PINN, Cf. merely decreasing the model size of PINNs leads to poor performance (Appendix S). Figure 4(a) shows how the model adaptively learns the rank structure for varying values of $\beta$. We observe that, with the low-rank format, Hyper-LR-PINN provides more than ($\times$4) speed up in training time compared to the vanilla PINN. More comparisons on this aspect will be presented in the general CDR settings.

**Ablation on fixed or learnable basis vectors:** We compare the performance of the cases, where the basis vectors $\{U_r^l, V_r^l\}$ are trainable. We observe that the fixed basis vectors lead to an order of magnitude more accurate predictions as well as faster convergence (See Appendix Q for plots).

**Ablation on orthogonality constraint:** We compare the performances of the cases, where the proposed model is trained without the orthogonality constraint, Eq. (2). Experimental results show that the orthogonality constraint is essential in achieve good performance (See Appendix J).

**Many-query settings (inter/extra-polation in the PDE parameter domain):** Next, we test our algorithm on the *many-query* scenario, where we train the proposed framework with the 11 training PDE parameters $\beta \in [30, 40]$ (Phase 1), freeze the hypernetwork and the learned basis, and retrain only the diagonal elements for 150 unseen test PDE parameters (Phase 2). Figure 5 depicts the absolute and relative errors of the solutions, which are as accurate as the ones on the training PDE parameters, in particular, for $\beta < 30$ (i.e., extrapolation). Although the model produces an increasing error for $\beta > 40$, the relative error is still around $\sim$10%, which cannot be achievable by vanilla PINNs. The right chart of Figure 5 shows the number of trainable model parameters, which is determined by the rank (i.e., the output of the hypernetwork). As PINNs fail to produce any reasonable approximation in this $\beta$ regime, we make comparisons in the general CDR settings below.

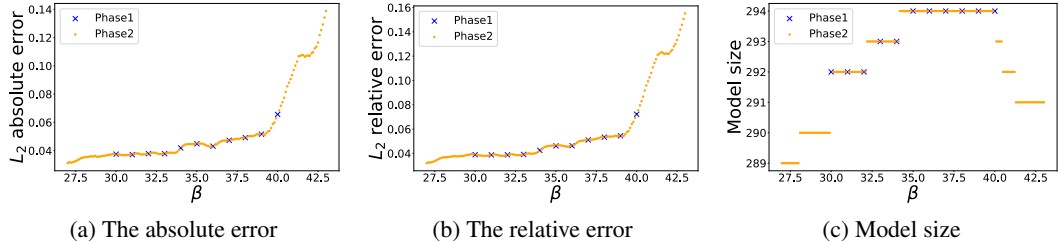

| (a) The absolute error | (b) The relative error | (c) Model size |

Figure 5: Multi-query scenario: the absolute/relative errors of the predicted solutions at unseen test parameters (left and middle), and the number of trainable model parameters (right).

Table 4: The relative errors of the solutions of parameterized PDEs with $\beta, \nu, \rho \in [1,5]$. The initial condition of each equation is Gaussian distribution $N(\pi, (\pi/2)^2)$.

| PDE type | No pre-training | | | Meta-learning | | |
|---|---|---|---|---|---|---|
| | PINN | PINN-P | PINN-S2S | MAML | Reptile | Ours |
| Convection | 0.0327 | 0.0217 | 0.2160 | 0.1036 | 0.0347 | 0.0085 |
| Reaction | 0.3907 | 0.2024 | 0.5907 | 0.0057 | 0.0064 | 0.0045 |
| Conv-Diff-Reac | 0.2210 | 0.2308 | 0.5983 | 0.0144 | 0.0701 | 0.0329 |

### 4.2.2 Performance on the general case of the CDR equation

**Solution accuracy:** Next, we present the performance of the proposed method on solving the benchmark PDEs (not only in the failure mode but) in all available settings for $\beta, \nu, \rho$. Due to space reasons, we present only a subset of our results and leave the detailed results in Appendix M. Table 4 shows that the proposed Hyper-LR-PINN mostly outperforms the baselines that operate in full rank.

**Some observations on learned low-rank structures:** Again, the proposed method learns rank structures that are adaptive to the PDE parameter (Figure 4(b)). The learned rank structures vary more dynamically ($r$ from 20 to 34) as the PDE parameter range [1,20] is larger compared to [30,40].

Figure 4(c) visualizes the magnitude of the learned diagonal elements $\boldsymbol{s}^2$ in the second hidden layer. The horizontal axis indicates the index of the element in $\boldsymbol{s}^2$, which is sorted in descending order for analysis, and the vertical axis indicates the $\beta$ value. The depicted result agrees with our intuitions in many aspects: i) the values decay fast (indicating there exist low-rank structures), ii) the value of each element $\boldsymbol{s}_i^2$ either increases or decreases gradually as we vary $\beta$, and iii) for higher $\beta$, higher frequency basis vectors are required to capture more oscillatory behavior exhibited in the solutions, which leads to higher ranks. We report the information for other layers and other problems in Appendix L.

**Computational cost:** Here, we compare the computational cost of Hyper-LR-PINN and PINN in a *many-query* scenario. We train Hyper-LR-PINN for $\beta \in [1, 20]$ with interval 1 (Phase 1) and perform Phase 2 on $\beta \in [1, 20]$ with interval 0.5 (including unseen test PDE parameters). We record the number of epochs required for PINN and Hyper-LR-PINN to reach an L2 absolute error of less than 0.05. Figure 6 reports the number of required epochs for each $\beta$, showing that the number of epochs required for Hyper-LR-PINN slowly increases while that for PINNs increases rapidly; for most cases, Hyper-LR-PINN reaches the target (i.e., 0.05) in only one epoch. We also emphasize that Hyper-LR-PINN in Phase 2 requires much less FLOPS in each epoch (See Section 3.2).

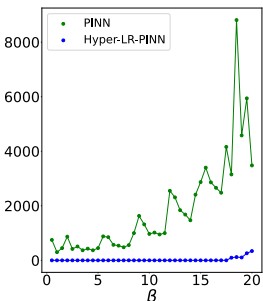

Figure 6: Computational cost in epochs

### 4.2.3 Performance on the 2D Helmholtz equation

Now we report experimental results on 2D Helmholtz equations for varying $a_1$ and $a_2$. Under the same condition in Table 2, we train Hyper-LR-PINN for $a \in [2, 3]$ with interval 0.1 ($a = a_1 = a_2$), and compare the performance with PINN. Surprisingly, Hyper-LR-PINN approximates the solution

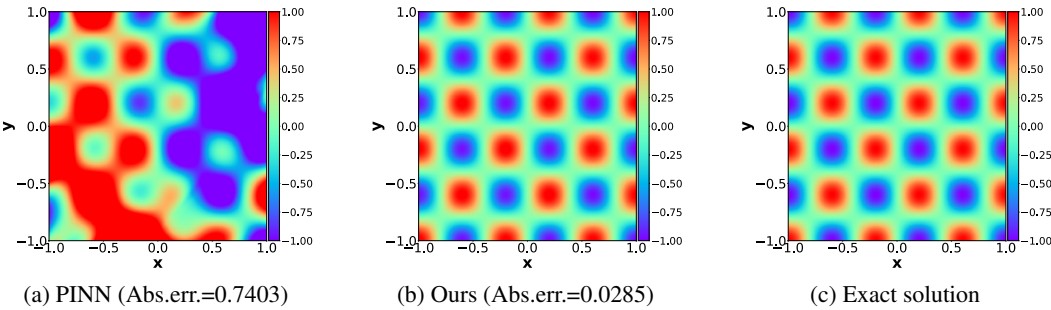

(a) PINN (Abs.err.=0.7403)  (b) Ours (Abs.err.=0.0285)  (c) Exact solution

Figure 7: [2D-Helmholtz equation] Solution snapshots for $a = 2.5$

with high precision in only 10 epochs in Phase 2, while PINN struggles to find accurate solutions over 2,000 epochs (Figure 7). We show more visualizations and detailed results in Appendix T.

## 5   Related Work

**Meta-learning of PINNs and INRs:**   HyperPINNs [22] share some commonalities with our proposed method in that they generate model parameters of PINNs via hypernetwork [23]. However, hyperPINNs can only generate full-rank weights and do not have capabilities to handle parameterized PDEs. Another relevant work is optimization-based meta-learning algorithms; representatively MAML [20] and Reptile [21]. In the area of INRs, meta-learning methods via MAML and Reptile for obtaining initial weights for INRs have been studied in [24]. In [25, 26], meta-learning methods, which are based on MAML, have been further extended to obtain sparse representation of INRs.

**Low-rank formats in neural networks:**   In natural language processing, the models being employed (e.g., Bert) typically have hundreds of millions of model parameters and, thus, making the computation efficient during the inference is one of the imminent issues. As a remedy, approximating layers in low-rank via truncated SVD has been studied [14, 15]. Modeling layers in low-rank in general has been studied for MLPs [27–30] and convolutional neural network architectures [31]. In the context of PINNs or INRs, there is no low-rank format has been investigated. The work that is the closest to our work is SVD-PINNs [32], which represents the hidden layers in the factored form as in Eq. (1), but always in full-rank.

**PINNs and their variants:**   There have been numerous sequels improving PINNs [1] in many different aspects. One of the main issues is the multi-term objectives; in [19], a special gradient-based optimizer that balances multiple objective terms has been proposed, and in [33], a network architecture that enforce boundary conditions by design has been studied. Another main issue is that training PINNs often fails on certain classes of PDEs (e.g., fail to capture sharp transition in solutions over the spatial/temporal domains). This is due to the spectral bias [34] and, as a remedy, in [18], training approaches that gradually increase the level of difficulties in training PINNs have been proposed. In [35], a Lagrangian-type reformulation of PINNs is proposed for convection-dominated PDEs. There are also other types of improvements including Bayesian-version of PINNs [36] and PINNs that enforce conservation laws [37]. All these methods, however, require training from scratch when the solution of a new PDE is sought.

## 6   Conclusion

In this paper, we propose a low-rank formatted physics-informed neural networks (PINNs) and a hypernetwork-based meta-learning algorithm to solve parameterized partial differential equations (PDEs). Our two-phase method learns a common set of basis vectors and adaptive rank structure for varying PDE parameters in Phase 1 and approximate the solutions for unseen PDE parameters by updating only the coefficients of the learned basis vectors in Phase 2. From the extensive numerical experiments, we have demonstrated that the proposed method outperforms baselines in terms of accuracy and computational/memory efficiency and does not suffer from the failure modes for various parameterized PDEs.

**Limitations:** Our Hyper-LR-PINN primarily concentrates on the cases where the PDE parameters are the coefficients of PDE terms. As a future study, we plan to extend our framework to more general settings, where the PDE parameters define initial/boundary conditions, or the shape of domains. See Appendix A for more details.

**Broader impacts:** Hyper-LR-PINN can solve many equations even for PINN's failure modes with high precision. However, despite its small errors, one should exercise caution when applying it to real-world critical applications.

## Acknowledgments

This work was supported by the Institute of Information & Communications Technology Planning & Evaluation (IITP) grant funded by the Korean government (MSIT) (No. 2020-0-01361, Artificial Intelligence Graduate School Program ay Yonsei University, 10%), and (No.2022-0-00857, Development of AI/databased financial/economic digital twin platform, 90%).

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
