# OpenReview forum: "Hypernetwork-based Meta-Learning for Low-Rank Physics-Informed Neural Networks"
_NeurIPS.cc/2023/Conference — NeurIPS 2023 spotlight_

### Official Review · Reviewer_wUrs · 2023-06-20

**Soundness:** 3 good
**Presentation:** 3 good
**Contribution:** 3 good
**Rating:** 7
**Confidence:** 2

**Summary:**

The paper presents a novel low-rank PINN architecture to solve parametrized partial differential equations (PDEs). Based on a hyper network framework they compute solutions to PDEs in a rank-adaptive way. Their approach is able to overcome for certain test systems known failure modes of PINNs. The method is extensively evaluated and benchmarked on various tasks for PDEs in 1D and 2D.

**Strengths:**

Disclaimer: I am not up to date with the most recent architectural improvements regarding PINNs and state-of-the-art accuracy.

To the best of my knowledge, the architectural improvements regarding the low-rank approximation are novel and the idea to predict the singular values with a hyper network is an interesting approach to learn a neural network to represent multiple parametrizations of PDEs:

1.	The method section is well written and easy to follow. All details are sufficiently explained, and the paper is well organized.
2.	The authors carried out extensive experiments to support their architectural changes and compare it against different methods.
3.	The results seem promising. Especially the results regarding the failure modes are impressive with an order of magnitude better accuracy (rel. and abs.) in Table 2 with significantly fewer parameters.


**Weaknesses:**

The results are impressive, as mentioned above, but to me it is not clear why the hyper network is so crucial to overcome failure cases:

1.	In my opinion the paper would benefit from a more prominent motivation, why the proposed approach is more robust against failure modes. Is it connect to the motivation in appendix D.2 and the remark in appendix B?
2.	In this regard an ablation on the architectural changes would also help to better understand the improvements proposed. To my understanding the closest PINN architectures is the HyperPINN method. Would it make sense to compare it against the newly proposed ansatz for example in Table 2, as another method w/o pre-training?
3.	In section 4.2.2 (“computational cost”) a fair comparison would be the meta-learning strategies, especially since some of the parametrizations of $\beta$ are in the training set (for Phase 1).


**Questions:**

1.	Why is your model more robust against the known “failure modes”? In Table 2 it seems a Naïve LR-PINN is not able to reach the same accuracy, but considering a full rank version, it should be as expressive as the Hyper-LR-PINN approach. Could you elaborate? (related to the first bullet point in the weakness section)
2.	Is it clear why in the general case of CDR equations (Table 4) the improvements diminish compared to the benchmark methods?
3.	Is the PINN-P approach also pre-trained on multiple parametrizations in appendix N? Judging from Table 5 it seems that PINN-P is randomly initialized, but wouldn’t it be possible to pre-train it? This could potentially also improve results on PINN-P.
4.	Overall, I had problems understanding how the datasets are decomposed, especially when comparing Hyper-LR-PINN with a vanilla PINN or variations (PINN-P). As far as I understood Hyper-LR-PINN is pre-trained on various parametrizations including the target parametrization and then in Phase 2 finetuned on the target. Does this mean Hyper-LR-PINN sees more datapoints on the target then the vanilla PINN?


**Limitations:**

The limitations are addressed in the main text and also the supplementary information.

---

> ### Author Rebuttal · Authors · 2023-08-09
>
> **Q1: In my opinion the paper would benefit from a more prominent motivation, why the proposed approach is more robust against failure modes. Is it connect to the motivation in appendix D.2 and the remark in appendix B?**
>
> **A1:** Your understanding of our motivation is correct; we were motivated by the observations in appendix D.2 and the remark in appendix B, which leads to development of the proposed approach. Although we empirically presented that the proposed algorithm is effective in solving the “failure modes”, we lack theoretical explanation on the phenomena we are observing.
> We conjecture that learning the set of basis vectors (close to orthogonal) that works well for a range of PDE parameter values seems to be the key aspect that enables the model to avoid spectral bias and, thus, facilitates approximating highly oscillatory (cf. convection equations) and very stiff (cf. reaction equations).
>
> \
> **Q2: In this regard an ablation on the architectural changes would also help to better understand the improvements proposed. To my understanding the closest PINN architectures is the HyperPINN method. Would it make sense to compare it against the newly proposed ansatz for example in Table 2, as another method w/o pre-training?**
>
> **A2:** We indeed have a result of applying HyperPINN on the “failure mode’’ of the convection equations, shown in Appendix S. This result suggests that the improved performance of Hyper-LR-PINN is more than just using hypernetwork. As elaborated in the response to the previous comment, we believe the training algorithm accounts more for the improved performance.
>
> \
> **Q3: In section 4.2.2 a fair comparison would be the meta-learning strategies, especially since some of the parametrizations of $\beta$ are in the training set (for Phase 1).**
>
> **A3:** Please see global comment #1.
>
> \
> **Q4: Why is your model more robust against the known “failure modes”? In Table 2 it seems a Naïve LR-PINN is not able to reach the same accuracy, but considering a full rank version, it should be as expressive as the Hyper-LR-PINN approach. Could you elaborate?**
>
> **A4:** For Naïve-LR-PINN, the orthogonal basis is obtained from the randomly initialized weights of the layers. In contrast, for Hyper-LR-PINN, the basis is learned from the weights of the layers that represent the parameterized PDEs. Consequently, Hyper-LR-PINN can obtain a basis that is well-suited for representing the target parameterized PDEs. This hypernetwork based meta-learning methodology provides a favorable starting point to the model before entering the Phase 2 (See appendix R). The experiments in this paper empirically demonstrate that the obtained basis can more easily generalize to PDE coefficients that were not trained during Phase 1, as compared to Naïve-LR-PINN.
>
> \
> **Q5: Is it clear why in the general case of CDR equations (Table 4) the improvements diminish compared to the benchmark methods?**
>
> **A5:** We believe that equations containing diffusion phenomena smooth out the solution snapshots and, thus, make the all considered methods somewhat comparable to each other. In such situations, PINNs in full-rank have a tendency to produce slightly better accuracy as they have higher expressivity. As shown in the additional comparisons (cf. Figure 1 of attached PDF), we also expect that the other competing baselines (MAML, Reptile) would have trouble finding good initializations for PINNs when the ranges of PDE coefficients become larger.
>
> \
> **Q6: Is the PINN-P approach also pre-trained on multiple parametrizations in appendix N? Judging from Table 5 it seems that PINN-P is randomly initialized, but wouldn’t it be possible to pre-train it? This could potentially also improve results on PINN-P.**
>
> **A6:** Please see global comment #2.
>
> \
> **Q7: Overall, I had problems understanding how the datasets are decomposed, especially when comparing Hyper-LR-PINN with a vanilla PINN or variations (PINN-P). As far as I understood Hyper-LR-PINN is pre-trained on various parametrizations including the target parametrization and then in Phase 2 finetuned on the target. Does this mean Hyper-LR-PINN sees more datapoints on the target then the vanilla PINN?**
>
> **A7:** Thanks for the comments. To address this concern, we’d like to emphasize that this is why we included baselines such as MAML and Reptile and put all those considered methods in the same experimental environments. Those are the PINN variants for solving parameterized PDEs and we can use the same setups for training/test data. All MAML,  Reptile, and the proposed Hyper-LR-PINN see the same number of coordinates (x,t) at the same number of PDE parameters ($\mu$) during the training.
>
> For PINN-P, thanks to the reviewer’s suggestion, we were able to perform the experiments on the same training and data setup, and reported the result in Tables 1 and 2 of the attached PDF.  With those additional experiments, we will add this description on the dataset in the manuscript to address the concern.

---

> > ### Comment · Reviewer_wUrs · 2023-08-16
> >
> > Thank you for your detailed response and the additional experiments! All my concerns were addressed and I will raise my score.

---

### Official Review · Reviewer_uxdN · 2023-07-04

**Soundness:** 3 good
**Presentation:** 3 good
**Contribution:** 3 good
**Rating:** 7
**Confidence:** 3

**Summary:**

In this paper, a meta-learning method is proposed for learning parameter-dependent dynamical systems. In particular, similarly to the reduced order method, the proposed method uses a low-rank neural network to construct a model, and the parameter-dependent coefficients are learned separately by a neural network. By combining these two, it is possible to retrieve the important modes, depending on the parameters of the system. This makes it possible to build models that perform well despite the small model size.

**Strengths:**

As far as I know, modeling differential equations by low-rank neural networks is certainly new. Based on the idea of the reduced order method, I suppose that this is a natural and reliable approach. The construction of a small model is also desirable for simulation and other applications. In addition, the paper is clearly written.

**Weaknesses:**

Although it may not be a weakness, it is difficult to understand theoretically that learning singular vectors does not improve performance when the matrix is decomposed into lower ranks. Based on the idea of model order reduction, the singular vectors should correspond to the important modes for expressing the solution and should be learned according to the data. In this regard, there appears to be a discrepancy between the idea and the results.

**Questions:**

Why does learning singular vectors not improve performance? I have an impression that this situation is similar to the reservoir computing, in which multiple random modes are first constructed and only a part of the parameters are adjusted to form a model. Should this method be understood as such a method rather than as a reduced order method?

**Limitations:**

No potential negative societal impact is expected.

---

> ### Author Rebuttal · Authors · 2023-08-09
>
> **Q1: Although it may not be a weakness, it is difficult to understand theoretically that learning singular vectors does not improve performance when the matrix is decomposed into lower ranks. Based on the idea of model order reduction, the singular vectors should correspond to the important modes for expressing the solution and should be learned according to the data. In this regard, there appears to be a discrepancy between the idea and the results.**
>
> **Why does learning singular vectors not improve performance? I have an impression that this situation is similar to the reservoir computing, in which multiple random modes are first constructed and only a part of the parameters are adjusted to form a model. Should this method be understood as such a method rather than as a reduced order method?**
>
>
> **A1:** Thank you for the comment. Indeed, this is a valid point; making the basis vector learnable in phase 2 would increase expressivity. We believe, however, the difficulty comes from the training algorithm, where the similar phenomena were observed in matrix decomposition methods. Consider finding a low-rank decomposition of matrix such that min $|| X - UDV^T ||_A$ with a norm induced from an inner product. In such a problem, updating $U,D,V$ all together in an interactive solver typically introduces more complexity. To avoid such difficulty, special solvers have been developed such as direct linear algebraic decompositions [Golub and Van Loan, 1996] which does not use a gradient-based optimization method, or notably, alternating minimization for matrix completion [Jain, et al, 2013].  In the context of low-rank approximation of solutions of PDEs, similar alternating approaches (e.g., alternating least-squares or alternating energy minimization) have been shown to be more effective [Doostan and Iaccarino, 2009], [Dolgov and Savostyanov, 2014].
>
> Developing such a specialized solver in the setting of low-rank MLP can be considered as out of scope. We will mention this aspect in the limitation section where we discuss potential future directions.
>
> We agree that there is a very high-level analogy that can be made to reservoir computing (RC), since in the second phase of our two-phase training procedure only part of the parameters are trained. However, this is a feature shared with all models making use of encoding-decoding schemes. Moreover, an important distinguishing feature in RC is the recurrent neural network, whereas our neural network architecture is feedforward. The fact that we are specifically solving parametrized partial differential equations also makes our model more specialized. We believe these points makes our model much more similar to a reduced order model, where a dimension-reduced implicit representation is learned in the first phase. In the second phase, a family of functions is quickly approximated by training a few parameters.
>
>
> [Golub and Van Loan, 1996]  Golub, Gene H. and Van Loan, Charles F., Matrix Computations, The Johns Hopkins University Press, 1996.
>
> [Jain, et al, 2013] Jain, P., Netrapalli, P., Sanghavi, S.: Low-rank matrix completion using alternating minimization. In: Proceedings of the forty-fifth annual ACM Symposium on Theory of Computing, pp. 665–674. ACM (2013)
>
> [Doostan and Iaccarino, 2009] Doostan, A., Iaccarino, G.: A least-squares approximation of partial differential equations with high-dimensional random inputs. J. Comput. Phys. 228(12), 4332–4345 (2009)
>
> [Dolgov and Savostyanov, 2014] Dolgov, S.V., Savostyanov, D.V.: Alternating minimal energy methods for linear systems in higher dimensions. SIAM J. Sci. Comput. 36(5), A2248–A2271 (2014)

---

> > ### Comment · Reviewer_uxdN · 2023-08-18
> >
> > Thank you very much indeed for the detailed explanation! My concern is fully addressed and I have raised my score.

---

### Official Review · Reviewer_eb2d · 2023-07-04

**Soundness:** 4 excellent
**Presentation:** 3 good
**Contribution:** 4 excellent
**Rating:** 8
**Confidence:** 3

**Summary:**

This work aims to improve on the efficiency and performance of adapting physics-informed neural networks (PINN) when dealing in many-query settings, given that the present field is limited in computationally-demanding retraining on new PDE parameters. It specifically focuses on learning low-rank versions of PINNs that enable low-parameter models through a hypernetwork-based meta-learning framework. A two-stage learning framework is proposed, in which 1) the hypernetwork and a low-rank PINN (LR-PINN) is trained across varying PDE parameters to learn a general set of basis vectors and 2) the basis vectors of the LR-PINN are fixed and the hypernetwork is used to generate an initial diagonal set on the novel PDE parameter configuration which are then trained on the new collocation points. In a set of experiments and ablations on challenging 1-D and 2-D PDEs that are known to have common "failure modes" for PINNs, the Hyper-LR-PINN demonstrated significant computation and performance improvements over both traditional PINN and meta-learning PINN baselines.

**Strengths:**

The blend of hypernetworks and meta-learning techniques in the PINN space in order to build efficient models that can work or adapt across a variety of PDE parameters is of increasing notice and importance. This work is positioned well against similar recent works and provides an intuitive and thought-out approach for handling all of these components together, especially the smart use of hypernetworks in outputting a smaller set of important parameters as a component of the low-rank decomposition.

- The writing is clean and clear and the sections are structured well.
- The algorithm is explained clearly and it is easy to follow the all the moving components.
- The research questions problems are clearly laid out and have corresponding answers in the model design.
- The experiments are robust, with ablations that support each of the research question, and a comprehensive suite of baselines are considered.
- The appendix is similarly robust, providing both visualizations and analogies to relevant background techniques that enhances understanding. The inclusion of the preliminary experiments that motivated the techniques proposed is additionally welcome.
- The code is clean and reproducible, given provided environment files and a detailed README with instruction. All data generators are provided as well.

**Weaknesses:**

- The lack of explanation on the meta-learning inclusion to the framework in the main paper is a bit confusing if a reader isn't coming from a meta-PINN perspective as there are no definitions given in the standard form of meta-train and meta-test environments or details on the context/query sets. It is elaborated on more in the Appendix and it can be pieced together in the work but perhaps a more clear reference to that section in the main work or elaboration on how meta-learning is formulated in this domain may provide more general clarity.

**Questions:**

** **

**Limitations:**

Societal impact and limitations of the model/experimentation is described appropriately, with further information on future work provided in the supplementary material.

---

> ### Author Rebuttal · Authors · 2023-08-09
>
> **Q1: The lack of explanation on the meta-learning inclusion to the framework in the main paper is a bit confusing if a reader isn't coming from a meta-PINN perspective as there are no definitions given in the standard form of meta-train and meta-test environments or details on the context/query sets. It is elaborated on more in the Appendix and it can be pieced together in the work but perhaps a more clear reference to that section in the main work or elaboration on how meta-learning is formulated in this domain may provide more general clarity.**
>
> **A1:** We appreciate the suggestion. We agree that readers who are not from a meta-PINN perspective may have confusion. Considering the page limitation, we think it would be better to put a clear reference to the sections in the main text. We will make this change in the manuscript at the camera-ready version.

---

### Official Review · Reviewer_W4Fc · 2023-07-07

**Soundness:** 3 good
**Presentation:** 3 good
**Contribution:** 3 good
**Rating:** 6
**Confidence:** 3

**Summary:**

The paper proposes an approach for solving PDEs by using a low-rank parameterized NN.
While traditional PINNs are restricted to a single PDE instance, the authors use a hypernetwork to be able to solve PDEs with different parameters. The hypernetwork predicts the weights matrices of a PINN network in a low-rank fashion: each weight matrix is decomposed via SVD, and only the singular values diagonal matrix are modulated by the hypernetwork, while the other two matrices are shared across all the different PDEs. This allows quick inference on unseen PDEs, by learning only the diagonal terms.
Performance is evaluated on 1D-convection-reaction-diffusion and 2D-Helmholtz PDEs. It is shown that this low-rank hypernetwork architecture allows to learn on previous failure cases of the vanilla PINN (eg high convection or high reaction).
The hypernetwork method shows strong performance, compared to other meta-learning methods (MAML, Reptile), as well as other PINN variants.

**Strengths:**

- The paper is well-written, and provides an extensive set of experiments and ablations showing the relevance of the low-rank decomposition, proposed loss terms and evaluate
- The performance of the proposed model is very competitive, while also providing much faster inference compared to the other meta-learning algorithms, and generalizing better than the other PINN variants

**Weaknesses:**

- “low-rank PINNs containing only hundreds of model parameters” (l. 10): the parameter comparison seems unfair, since only the parameters learned during phase 2 are counted (not hypernetwork or basis vectors parameters)
- The adaptive rank (section K) in appendix is missing

**Questions:**

- "For higher values (20), it appears that a higher rank is required to achieve a certain level of accuracy, presumably due to numerical reasons.” (l.503): what do you mean by numerical reasons?
- How does the proposed architecture scale to higher dimensions?
- How does the Phase 1 computation compare to the PINN variants?

**Limitations:**

The method is currently limited to generalizing to new PDE parameters only, it would be interesting to see if it can be extended to different initial / boundary conditions

---

> ### Author Rebuttal · Authors · 2023-08-09
>
> **Q1: “low-rank PINNs containing only hundreds of model parameters” (l. 10): the parameter comparison seems unfair, since only the parameters learned during phase 2 are counted (not hypernetwork or basis vectors parameters)**
>
> **A1:** Thank you for the comment. We will make the statement more precise to be fair with baselines. Our planned edit would be as follows:
>
> “To address this issue, we propose lightweight low-rank PINNs containing only hundreds of model parameters, enabled by meta-learning a set of common basis vectors for weights and an associated hypernetwork, which allow efficient solution approximations for varying PDE input parameters.”
>
> \
> **Q2: The adaptive rank (section K) in appendix is missing**
>
> **A2:** Thank you for pointing this out. The content of Section K was presented not in an ideal place, causing the confusion. We reorganized the sections in Appendix for better legibility.
>
>
> \
> **Q3: "For higher values (20), it appears that a higher rank is required to achieve a certain level of accuracy, presumably due to numerical reasons.” (l.503): what do you mean by numerical reasons?**
>
> **A3:** We wrote "numerical reasons" to mean that, while there is a theoretically optimal approximation that guarantees equivalent error rates for different parameter values, there is no known learning procedure that finds this optimal approximation numerically at equivalent rates of convergence. That is to say, the learning problem we consider here is known to have more challenging and non-convex loss landscapes for optimization for higher parameter values [18]. Therefore, we speculated that representing these more complex phenomena would require a higher rank, and we observed such behavior in experiments, as shown in Figure 4 of the main paper.
>
> In the model reduction or numerical analysis literature, here a heuristic argument based on numerical experiments that suggest a trade-off between the rank and the ease of numerical approximation [Welper, 2017] unless some nonlinear transformation is applied [Rim et al, 2018].
>
> We plan to edit the text to make this more clear, as follows:
>
> "For higher values (20), it appears that a higher rank is required to achieve a certain level of accuracy, presumably due to the difficulty of the numerical optimization problem."
>
> [18] Aditi Krishnapriyan, Amir Gholami, Shandian Zhe, Robert Kirby, and Michael W Mahoney. Characterizing possible failure modes in physics-informed neural networks. Advances in NeuralInformation Processing Systems, 34:26548–26560, 2021
>
> [Welper, 2017] G. Welper, Interpolation of Functions with Parameter Dependent Jumps by Transformed Snapshots, SIAM Journal on Scientific Computing 2017 39:4, A1225-A1250
>
> [Rim et al 2023] D. Rim, B. Peherstorfer, and K.T. Mandli, Manifold Approximations via Transported Subspaces: Model Reduction for Transport-Dominated Problems, SIAM Journal on Scientific Computing 2023 45:1, A170-A199
>
> \
> **Q4: How does the proposed architecture scale to higher dimensions?**
>
> **A4:** For higher dimensional PDEs, we expect the specification of our LR-PINN architecture should match with the PINN architectures that are shown to be effective in approximating solutions (for example, PINNs for 1D/2D/3D PDEs in the original PINN paper). They employ relatively small-size neural networks (e.g., an MLP with 9 layers with 20 neurons) and we expect that employing a similar architecture in a low-rank format would allow approximation of solutions with reasonable accuracies. In our main paper, we used the same model architecture to learn both the 1D Convection-Diffusion-Reaction (CDR) equations and the 2D Helmholtz equation, and successfully trained the equations in both cases. Also, we expect the size of the hypernetwork can be kept relatively small as we interpret that the hypernetwork plays a role of an interpolation function. The effectiveness of lightweight hypernetworks has been demonstrated in the experiments in the manuscript.
>
> \
> **Q5: How does the Phase 1 computation compare to the PINN variants?**
>
> **A5:** Please see global comment #1.

---

> > ### Comment · Reviewer_W4Fc · 2023-08-21
> >
> > Thank you for the explanations, I will keep my score

---

### Author Rebuttal · Authors · 2023-08-09

**Global comment #1: [Additional results of computational cost]**

Assuming that we solve PDEs for the same number of PDE parameters without our method and other baselines, the computation needed for each forward pass in Phase 1 is slightly higher than that of training all individual vanilla PINNs because the proposed method involves training the hypernetwork. At the cost of marginally increased computation, the proposed method exhibits many benefits; most prominently (1) for test PDE parameters, an error level of about 5% can be achieved within few epochs (See Figure 1 of the attached PDF) while the traditional method requires several thousand epochs, and (2) we resolve the difficulties that PINNs have in the failure modes.

Regarding the benefit (1) mentioned above, we conducted additional analyses on computational cost in a many-query scenario  for four models, PINN, MAML, Reptile, and Hyper-LR-PINN, for fair comparison. In the experiments, we trained the models for convection equations on various $\beta$ ranges, i.e., [1, 5], [1, 10], and [1, 20], and measured the number of epochs required to reach an L2 error less than 0.05. We trained three meta-learning models, MAML, Reptile, and Hyper-LR-PINN, for the convection equations with $\beta$ values with an interval of 1 in Phase 1, whereas testing the models with an interval 0.5 in Phase 2, i.e., the trained and tested $\beta$ settings do not overlap. The experimental results are summarized in Figure 1 in the attached PDF.

According to Figure 1, when learning convection equations with $\beta \in$ [1, 20] and $\beta \in$ [1, 10], the performance of two meta-learning baselines is observed to be inferior to PINN models without pre-training. Even for some beta values, the minimum threshold, i.e., an error level of about 5%, was not reached within 10,000 epochs so that they were not plotted. However, with $\beta \in$ [1, 5], the meta-learning baselines demonstrate superior performance compared to PINN in all beta values. On top of that, Hyper-LR-PINN stands out as the most effective approach, achieving the target with the least computational cost across all ranges. These experimental results indicate that existing meta-learning methodologies seem to have difficulties in finding a good “global/common” initialization for PINNs as the range of the PDE coefficients becomes larger. On the other hand, Hyper-LR-PINNs do not suffer from the same issue and are shown to be very effective over a wider range of PDE coefficients compared to existing meta-learning methods. We will include additional experimental results and discussions in the paper.

\
**Global comment #2: [Additional ablation studies on PINN-P]**

We have verified that PINN-P is amenable to pre-training since it models the parameterized solution. Following reviewers’ suggestion, we integrated an additional pre-training phase into PINN-P and conducted new experiments on convection equations with $\beta$ = {30, 35, 40}, which are PINN’s failure modes (cf. Tables 2 and 7). The fine-tuned results of PINN-P are summarized in Tables 1 and 2 from the attached PDF.

When training PINN-P, we followed the same experimental setups of experiments in Table 2. To be specific, in phase 1, we trained PINN-P using convection equations with $\beta \in$ [30, 40], and then fine-tuned the models for six specific convection equations where $\beta$ = {30, 35, 40} and $\beta$ = {41, 42, 43} (within and outside the learning range of phase 1, respectively) in phase 2, employing the pre-trained model from the phase 1. As shown in Tables 1 and 2 of the attached PDF, PINN-P with pre-training phase shows overall improvements compared to previous PINN-P experiments in Table 7. However, there still exists a substantial performance gap between PINN-P and Hyper-LR-PINN (Full rank, Adaptive rank). Both PINN-P and Hyper-LR-PINN demonstrated that the pre-training process can be beneficial in improving model performance. We will conduct further discussions on the pre-trained PINN-P model.

---

### Decision · Program_Chairs · 2023-09-21

**Decision:**

Accept (spotlight)

**Comment:**

In this paper, a meta-learning framework based on the Hypernetwork is employed to propose an approach for solving PDEs using a low-rank parameterized Physics-Informed Neural Network (LR-PINN). Through a series of experiments involving 1D and 2D PDEs, significant improvements in computation and performance were demonstrated over both conventional PINNs and meta-learning PINNs as baselines. Modeling differential equations with a low-rank neural network is indeed novel. This achievement holds a strong position in comparison to recent similar research. It can be regarded as a crucial contribution that enhances both the efficiency and performance of PINNs.